

# The departure of sperm whales (*Physeter macrocephalus*) in response to the declining jumbo squid (*Dosidicus gigas*) population in the central portion of the Gulf of California

Héctor Pérez-Puig[1,*], Alejandro Arias Del Razo[2,*], Daniela Ahuatzin Gallardo[2] and Jaime Bolaños[3,4]

[1] Marine Mammal Program, Prescott College Kino Bay Center for Cultural and Ecological Studies, Bahía de Kino, Sonora, Mexico
[2] Chemical & Biological Sciences, Universidad de las Américas Puebla, San Andrés Cholula, Puebla, México
[3] Instituto de Ciencias Marinas y Pesquerías, Universidad Veracruzana, Veracruz, Veracruz, México
[4] Caribbean-Wide Orca Project (CWOP), Cagua, Aragua, Venezuela
[*] These authors contributed equally to this work.

Corresponding authors
Héctor Pérez-Puig,
hector.perez@prescott.edu
Alejandro Arias Del Razo, alejandro.arias@udlap.mx

## ABSTRACT

As sperm whales are important predators that control energy flux in the oceans, changes in their population can be used as a sentinel to measure of ecosystem health. The present study conducted a sperm whale survey of the eastern Midriff Islands Region in the Gulf of California over the course of nine years, recording sightings and collecting photographs of the fluke of sperm whale individuals. A photo-identification catalog was compiled, while individual recapture data were used to estimate the population size in the central portion of the Gulf of California, using a Jolly-Seber POPAN open population model. The results obtained show a yearly population of between 20 and 167 sperm whales, with a super population of 354 sperm whales observed between 2009 and 2015. However, from 2016 to 2018, no sightings of the species were recorded, which coincides with the decline observed in landings of their main prey, the jumbo squid, in the region. General additive model conducted on sperm whale sightings per unit of effort vs jumbo squid landings obtained an adjusted $R^2$ of 0.644 and a deviance explained of 60.3%, indicating a good non-linear relationship between sightings of this odontocete and its prey availability. This evidence suggests that sperm whales departed the region between 2016 and 2018, due to a documented fishery collapse alongside changes of their main prey into its small phenotype, possibly as the result of increase warming conditions in surface and subsurface waters in the Gulf of California in the last three decades.

## INTRODUCTION

Sperm whales (*Physeter macrocephalus*) have been observed year-round, particularly during spring and summer, in the Gulf of California (GOC), suggesting that they may either comprise a resident population or present limited movements beyond its confines (*Gendron, 2000*). In the GOC, the relationship between sperm whale and its main prey, the jumbo squid (*Dosidicus gigas*), has been widely studied, revealing an overlapping distribution and similar dive-depth and habitat preferences (*Davis et al., 2007*; *Gallo-Reynoso, Égido-Villarreal & Coria-Galindo, 2009*). Moreover, it has been found that the jumbo squid and sperm whales occurring in the GOC share C and N stable isotope abundance (*Ruiz Castro, 2002*), showing that they share a close trophic relationship. In the GOC, the relationship between environmental variables, such as sea surface temperature and chlorophyll concentration, and sperm whale distribution has proven to be unrelated (*Gendron, 2000*), indicating that the presence of the sperm whale in the region may be more linked to prey availability.

Studies conducted during the 1990s linked variations in sperm whale movement patterns to *D. gigas* abundance (*Jaquet, Gendron & Coakes, 2003*), showing that sperm whales change their distribution based on the density of the jumbo squid population, but without leaving the GOC (*Jaquet & Gendron, 2002*). Previous studies support the hypothesis that the sperm whale's spatial distribution and feeding habits respond to small-scale resource availability (*Flinn et al., 2002*; *Whitehead et al., 2008*). However, it has also been suggested that, when prey resources are critically scarce, the population might undertake a large-scale movement in the face of the cost of staying in areas with low levels of food availability (*Whitehead, 2000*). For instance, studies carried out in the Galapagos archipelago have shown that sperm whale population density has varied drastically over the course of recent decades due to the fluctuation in the abundance of their prey, caused by climatological variations such as *El Niño* Southern Oscillations (ENSO) rather than mortality events in the sperm whale population (*Cantor et al., 2017*).

The jumbo squid was one of the most important commercial fishery in the GOC (particularly in the central portion) and even in all Mexico (*Nevárez Martínez et al., 2002*). Its population has suffered several documented fluctuations since the start of the species' commercial use in the 1970s. As jumbo squid feed on the mesopelagic micronekton that vary annually in the region, they respond rapidly to environmental changes, which is why their population may rapidly vary in density (*Markaida & Sosa-Nishizaki, 2003*). During the years of *La Niña* events (1996–1997), squid catches in the Guaymas Basin, Sonora, reached over 100,000 tons per year (*Markaida & Sosa-Nishizaki, 2001*); however, in the immediate aftermath of the 1997–1998 *El Niño* event, catches were recorded at a minimum of 3,000 tons (*Nevárez Martínez et al., 2002*). The population partially recovered from this event, with their maturing size increasing progressively (*Markaida, 2006*). In the last decade, the annual jumbo squid catch on the Sonoran central coast of the GOC reached almost 26,916 tons in 2009 and fell to its lowest point in 2018, when the annual catch for the Sonoran region was 169 tons (*CONAPESCA, 2018*), documenting this as a sustained
decline of the jumbo squid fishery in the GOC since 2009 and its imminent collapse in 2015 (*Frawley et al., 2019*).

Despite the efforts made to identify individuals that comprise the sperm whale population in the GOC, little is known about the factors that have influenced their population trend in recent years. From 1991 to 2010, *Ruvalcaba Márquez (2013)* photo-identified 589 individuals in an area between San Pedro Mártir and San José islands, while *Jaquet, Gendron & Coakes (2003)* photo-identified 159 individuals from 1998 to 1999, without reaching a plateau in their discovery curves, suggesting a significant portion of the population had not been photo-identified. Other studies that used photo-identification in 1996 and from 1998–2003 reported 567 (*Gerrodette & Palacios, 1996*) and 503 (*Jaquet & Gendron, 2004*) sperm whales, respectively. Studies based on mark-recapture models estimated that 1,070 individuals (95% CI =734-1623) use the area between La Paz Bay and San Pedro Mártir Island in the western part of the GOC (*Guerrero, Urbán & Rojas, 2006*), while another study that used the Petersen index yielded an estimate of 941 individuals (95% CI = 386-2,493; *Jaquet & Gendron, 2002*). In terms of the structure of the population, eight social units have been identified in the GOC (*Ruvalcaba Márquez, 2013*); however, no studies have focused on either the relationship between sperm whale sightings in the Midriff Islands Region of the GOC and jumbo squid landings or the size of the sperm whale population in the GOC in the last two decades.

The present study aims to estimate the population of sperm whales, as individuals occurring in the study area, in the eastern Midriff Islands Region of the GOC, using open population models to obtain the seasonal variation from 2010 to 2018, and provide new insight into the sperm whales occurring in the study area in the GOC regarding its relationship with the jumbo squid abundance, its main prey in the region.

## MATERIAL AND METHODS
### Study area
The Gulf of California (GOC) is located in the northwest of Mexico, with the Baja California Peninsula to the west and the states of Sonora and Sinaloa to the east and connected to the Pacific Ocean to the south. The GOC is 1,400 km long and its inner region is between 150 and 200 km wide and comprising approximately 900 islands and islets (*Lavín & Marinone, 2003*). It is well-known for its high productivity and biological biodiversity (*Alvarez-Borrego & Lara-Lara, 1991*) with two distinctive productive periods, comprising higher productivity in cold season (December to May) and lower productivity in warm season (July to October) (*Santamaría del Ángel, Álvarez-Borrego & Muller-Karger, 1994*; *Mardones, Marioni & Sierra, 1999*).

The Midriff Islands Region is located in the north-central part of the GOC characterized by a high density of islands, most notably the biggest ones, such as Ángel de la Guarda and Tiburon islands (Fig. 1) The Midriff Islands Region extends from the north of Ángel de la Guarda Island (29.56667°N, 113.57323°W), to San Pedro Mártir Island (28.38333°N, 112.30676°W) in the south and has basins that reach 900 m or up to 1,400 m deep, such as the Delfín and Salsipuedes Basins, respectively. This is one of the most outstanding

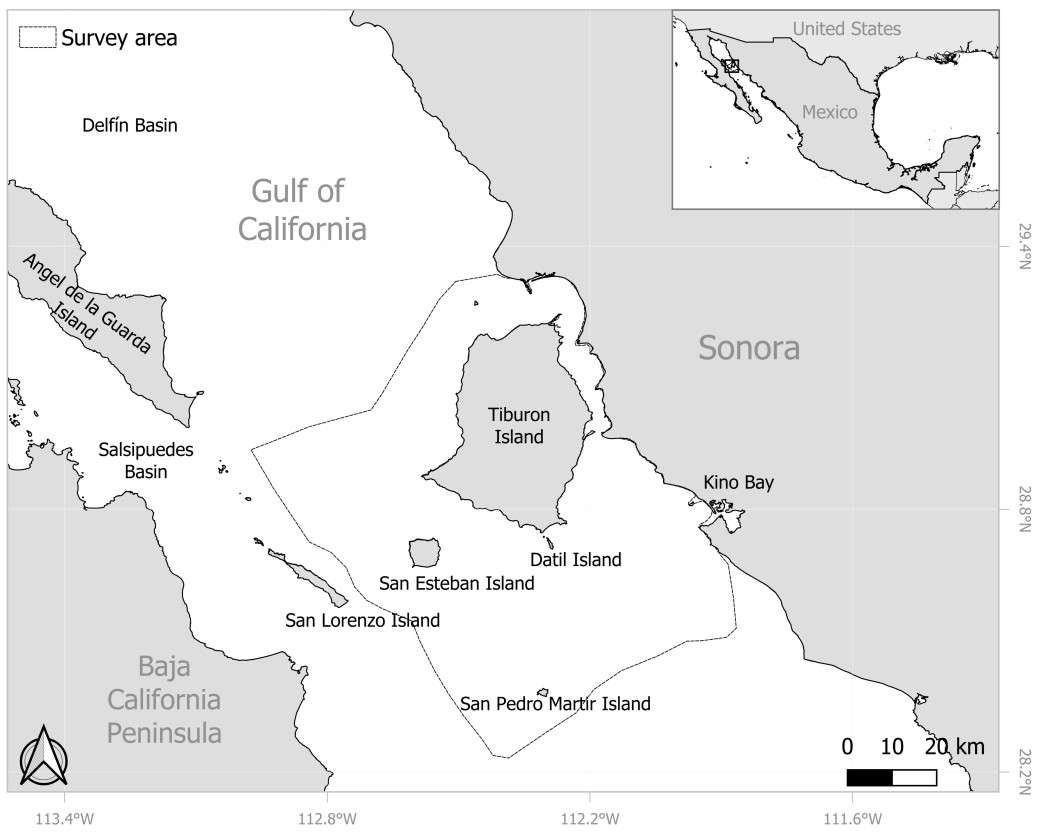

**Figure 1  Location of the eastern Midriff Islands Region and the central coast of Sonora in the Gulf of California.** Survey area around the eastern Midriff Island Region of the Gulf of California. Base map obtained from GADM (https://gadm.org/, 2024).

topographic features of the region and frames a unique oceanographic regime. The basins function as funnels and restrict circulation between the northern Gulf and the rest of the GOC; that generate an intense mixing of water due to strong tidal currents, causing upwelling areas that occur throughout the year (*Lavín & Marinone, 2003*; *Lluch-Cota, 2000*). This phenomenon provides a high marine productivity, making the region a priority site for the conservation of coastal and oceanic environments (*CONABIO, 2008*).

## Data collection

Records of sperm whales were obtained from surveys undertaken from January 2010 to November 2018, where a period spanned from December of one year to November of the next (hereinafter referred to as a year), considering the climatic conditions within the GOC. These surveys were collected under the SEMARNAT SGPA/DGVS/01329/12 permit and formed part of the cetacean studies conducted by the Marine Mammal Program of the Prescott College Kino Bay Center for Cultural and Ecological Studies in the area between adjacent waters of San Esteban Island and the central coast of Sonora hereinafter referred to as the eastern Midriff Islands Region of the GOC (approx. 5,700 km$^2$; Fig. 1). Four non-systematic surveys were made per month, navigate aboard a seven-meter-long

skiff powered by a 115-hp Yamaha outboard motor from Kino Bay at approximately 15 km/h, attempting to cover most of the study area, prioritizing areas with deep waters (>200 m) around Isla San Esteban and Isla San Pedro Mártir, the latter mentioned in previous works as an important aggregation area of sperm whales in the region (*e.g.*, *Jaquet and Gendron, 2009*). During surveys stops of no more than 20 min could be made regularly for omnidirectional scans in search of animals. The extent of the surveys varied depending on whether the meteorological conditions were favorable, always trying to navigate on a Beaufort sea state scale ≤ 3 and with minimal or no presence of swell. The survey effort was maintained for almost the entire year, except for the months of August (due to poor weather conditions and logistical limitations); however, this was consistent throughout all the years, with no statistically significant differences found between years (Kruskal-Wallis, $H = 12.56$, $n = 330$, $df = 8$, $p = 0.12$), At least two observers conducted each survey, performing continuous scans with the naked eye and/or using $8 \times 42$ mm Vortex binoculars to detect the sperm whale individuals. When a sperm whale sighting was recorded, the speed of the boat was reduced, and a slow and non-invasive approach was made. Each sighting was recorded on a field data sheet, with the data recorded including the number of individuals observed and the general behavior, while their geographical location was ascertained using a Garmin eTrex 30 GPS device (Global Positioning System) at first sight. Photographs of the sperm whale's fluke were taken using a digital Canon EOS 7D camera fitted with a 75–300 mm ultrasonic lens. The photographs were used to photo-identify all individuals observed at each sighting by comparing the unique natural marks (notches, scars, or missing parts) on their flukes, following established cetacean protocols (*Hammond, Mizroch & Donovan, 1990*; *Urian et al., 2015*; *Urian & Wells, 1996*). Once the individuals had been photo-identified, photo-recaptures within (intra-annual) and between (inter-annual) years were obtained using the ACDSee Photo Studio Ultimate 2020 software.

## Population estimation

The Jolly-Seber POPAN open population model was applied on all individuals that were captured and then recaptured on subsequent occasions, in order to estimate both the population by year and the super population, a parameter that estimates the total number of animals that pass through the study area during the whole period of study (*White & Burnham, 1999*). The U-CARE software (version 3.3; *Choquet et al., 2009*; *Choquet et al., 2005*) was used to assess the goodness of fit of the model to the data. The MARK software, version 8.2, was used to calculate these estimators using the maximum likelihood method (*White & Burnham, 1999*), which provides asymptotically-unbiased estimators, with a normal distribution and minimum variance, attaining a good level of precision in the estimates they generate, especially when a large number of individuals is photo-identified (*Cooch & White, 2023*; *Lebreton et al., 1992*; *Nichols, 2005*). Different models were tested using different parameters of probability of capture ($p_i$), apparent survival ($\phi_i$), and probability of entry into the population (*pent*), as either variable ($t$) or constant (.) over time. Estimates for the $p_i$, $\phi_i$, *pent*, and N parameters were calculated using maximum likelihood. From the resulting models, the most parsimonious model was selected using

the Akaike information criterion (quasi-likelihood AIC$_c$; *Akaike, 1998*), wherein the model with the lowest AIC$_c$ score best fit the data (*Burnham & Anderson, 2002*; *Cooch & White, 2023*; *Lebreton et al., 1992*).

**Sperm whale-jumbo squid correlation**

A large-scale fishery for jumbo squid (*Dosidicus gigas*) was established in the GOC in 1979 due to its high biomass (*Ehrhardt et al., 1982*). In this fishery, several periods of very low catches, collapses and periods of recovery have been documented between 1982 and 1999 (*Nevárez-Martínez et al., 2000*; *Morales-Bojórquez et al., 2001*) and recently. Which means that the intensive jumbo squid fishery in the GOC provides us with a unique opportunity to relate commercial squid landings data with the trend in sperm whale abundance. Jumbo squid landings data for the state of Sonora were obtained from the publicly-available databases maintained by the Mexican fishing authority (*CONAPESCA, 2018*). A general additive model (GAM) was used to ascertain the correlation between jumbo squid landings, as the independent variable, and sperm whale records calculated in terms of the individual sighted per unit effort (SPUE) value as the dependent variable, using the open-source software R (*R Core Team, 2023*) using libraries mgcv (*Wood, 2011*; *Wood, Pya & Saefken, 2016*) and ggplot2 (*Wickham, 2016*). SPUE was calculated by dividing the number of sperm whale sights per hours of survey. Code and data is available to download from https://zenodo.org/doi/10.5281/zenodo.13376738.

## RESULTS

From 2009 to 2018, nine years were conducted, comprising a total of 330 surveys and 2,190 h of survey effort and an average of 6.6 ($\pm$ SE 2.55) hours per survey, while the total number of hours of effort expended per year was between 186 and 273 h of effort. From 2009 to 2015, 73 sperm whale sightings were recorded, in which a total of 648 individuals were recorded: 398 adults; 21 juveniles; 47 calves; and 182 unclassified individuals. From 2016 to 2018, no sperm whale sightings were recorded (Fig. 2).

Using de photo-identification technique a total of 207 sperm whale individuals were photo-identified, of which 167 were identified as unique individuals, 15 were recaptured across multiple years, 18 and 22 were intra and inter-annually recaptured, respectively (Table 1). Of the 167 unique individuals identified by the present study, 149 ($\sim$88%) were seen only once and the remaining 12% were observed across more than one year, with only three individuals sighted across three or more years.

The goodness of fit test indicated compliance of the model with the data (Ch $i^2 = 3.33$, $df = 5$, $p = 0.65$). Formulations with pent time-constant showed either problems of convergence or a much higher AICc (of up two orders of magnitude) and where not included in our analysis. The most parsimonious model included time-variable probability of capture, survival and probability of entrance. This model presented null standard error for all parameters, so it was dropped down from the analysis (*Cooch & White, 2023*, p. 4.52). The first and last models showed a low Delta AICc (2.19) so we used the model averaging function to get the estimates of annual abundances.
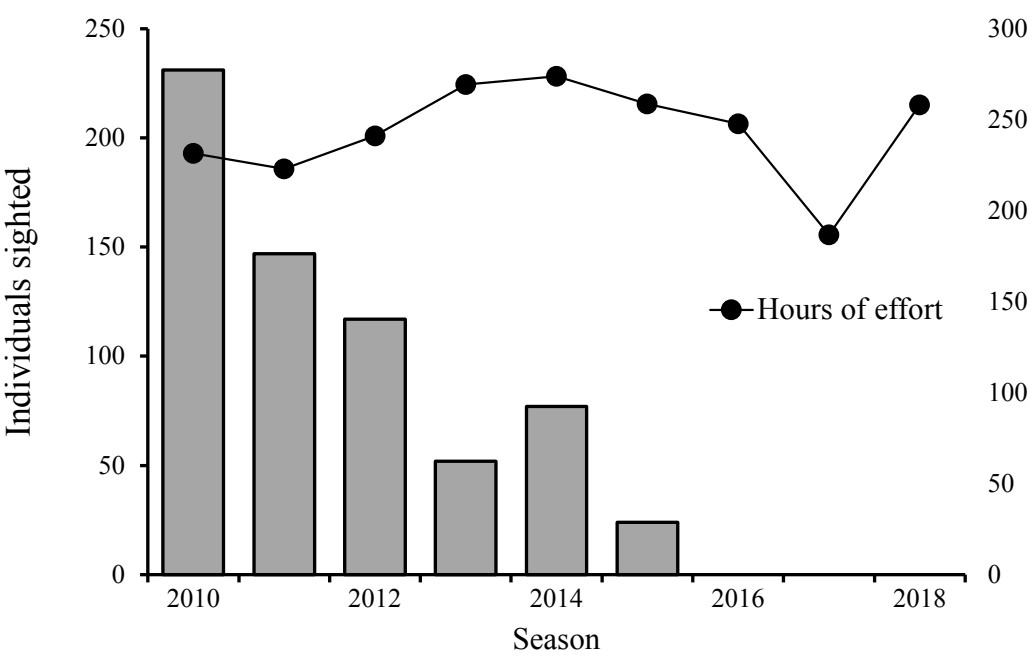

**Figure 2** **Total number of sperm whale sightings *vs* hours of effort per year.** Bars indicate the number of individual sperm whales sighted and the black dots the hours of survey effort for the same year.

**Table 1** **Summary of intra-annual and inter-annual recaptures of sperm whales (*Physeter macrocephalus*) in the survey years conducted from 2010–2018.**

| Year | Photo-identified | Intra-annual recaptures | Unique individuals | Inter-annual recaptures | Total unique individuals |
|---|---|---|---|---|---|
| 2009–2010 | 44 | 0 | 44 | 0 | 44 |
| 2010–2011 | 63 | 1 | 62 | 3 | 59 |
| 2011–2012 | 26 | 5 | 21 | 3 | 18 |
| 2012–2013 | 30 | 7 | 23 | 8 | 15 |
| 2013–2014 | 33 | 3 | 30 | 3 | 27 |
| 2014–2015 | 11 | 2 | 9 | 5 | 5 |
| 2015–2016 | 0 | 0 | 0 | 0 | 0 |
| 2016–2017 | 0 | 0 | 0 | 0 | 0 |
| 2017–2018 | 0 | 0 | 0 | 0 | 0 |
| Total | 207 | 18 | 189 | 22 | 167 |

The population estimation per year varied from 20 ($\pm$ SE 8.79) in the 2014-2015 year to a maximum of 167 ($\pm$ SE 70.82) in the 2010–2011-year, although it was not possible to estimate the population for the subsequent years (2015–2016, 2016–2017, and 2017–2018) because no sightings were recorded during them (Table 2). Also, from this model it was estimated that a super population of $\hat{N} = 354$ ($\pm95\%$, SE $=94.7$) sperm whales occurred through the study area between 2009 and 2015.

**Table 2** Estimated abundance of sperm whales (*Physeter macrocephalus*) in the eastern Midriff Islands Region of the GOC using an open population model [Phi(.) p(.) pent(t)] for the data corresponding to the 2009-2010 year up to the 2014–2015 year.

| Season | $\hat{N}$ | SE |
|---|---|---|
| 2009–2010 | 92 | 32.26 |
| 2010–2011 | 167 | 70.82 |
| 2011–2012 | 75 | 22.43 |
| 2012–2013 | 40 | 15.04 |
| 2013–2014 | 73 | 30.01 |
| 2014–2015 | 20 | 8.73 |

**Table 3** Squid landings in the waters off the state of Sonora (*CONAPESCA, 2018*), sperm whale sightings in the eastern Midriff Islands Region, and hours of survey effort.

| Season | Annual squid landings (thousands of tons) | Number of sperm whales sighted | Total hours of effort | Sights of sperm whales per unit of effort (SPUE; individuals sighted / hour) |
|---|---|---|---|---|
| 2009–2010 | 11.964 | 231 | 231.54 | 0.997 |
| 2010–2011 | 8.898 | 147 | 223.08 | 0.658 |
| 2011–2012 | 0.877 | 117 | 241.06 | 0.485 |
| 2012–2013 | 1.477 | 52 | 269.35 | 0.193 |
| 2013–2014 | 0.312 | 77 | 273.96 | 0.281 |
| 2014–2015 | 0.747 | 24 | 258.75 | 0.092 |
| 2015–2016 | 0.327 | 0 | 247.66 | 0 |
| 2016–2017 | 0.121 | 0 | 186.82 | 0 |
| 2017–2018 | 0.084 | 0 | 258.21 | 0 |

A considerable decline in jumbo squid (*Dosidicus gigas*) landings was recorded, falling from 26,916 (2009) to 139 (2018) tons, while sperm whale sighted fell from 231 individuals in the 2009–2010 year to 0 in the last three years of this study (2015-2016, 2016–2017, and 2017–2018), despite a sustained survey effort (Table 3). The number of sperm whale sighted per year were standardized to sighted per unit effort (SPUE). A general additive model (GAM) between the SPUE and jumbo squid landings (independent variable), using a quasi-Poisson family, showed a non-linear significant relationship ($p = 0.003$) with an adjusted $R^2$ of 0.644 and a deviance explained of 60.3%, indicating a good fit of the model (Fig. 3). This finding may suggest a strong correlation between the abundance of the sperm whale's main prey and its presence in the eastern Midriff Islands Region of the GOC.

## DISCUSSION

The GOC has gone through important environmental and ecological changes in the last 10–20 years (*Gilly et al., 2022*). An analysis of SST anomalies from a 100-year means

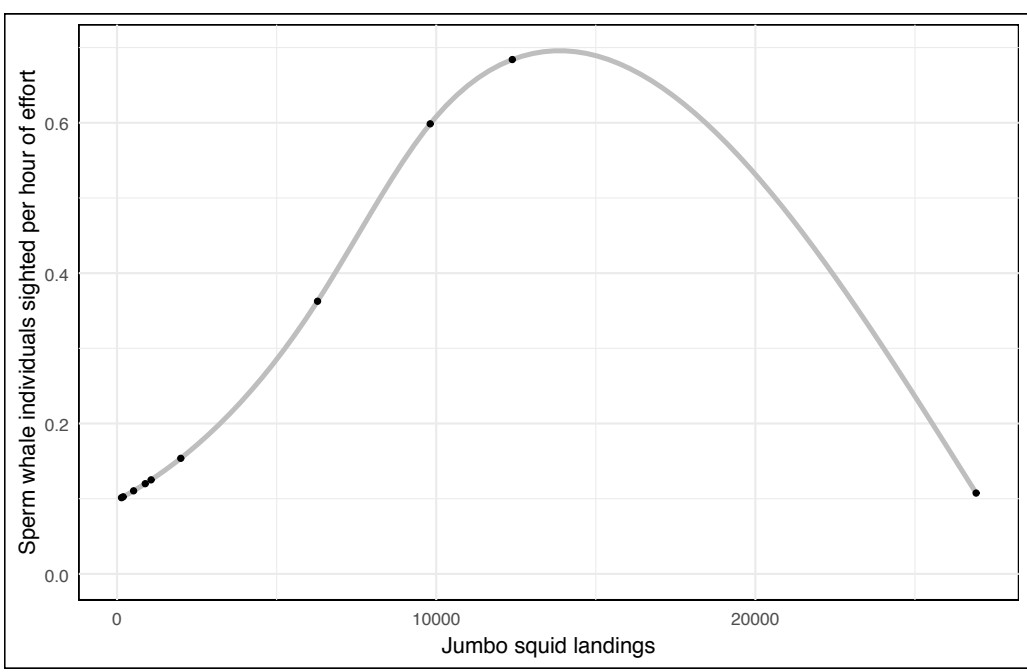

**Figure 3** **General additive model for sperm whale sightings per hour of effort *vs* jumbo squid landings.**
GAM shows a non-linear significant relationship ($p = 0.003$) between the sperm whales sighted per hour
of effort in the eastern Midriff Islands Region of the GOC and jumbo squid landings in the state of Sonora
(*CONAPESCA, 2018*).

between 1920 to 2020, showed a steady increase in temperature since the mid-1930s
and positive anomalies starting during the 1990s, which reach +1 °C by 2020 (*Adame
et al., 2020*). Remote sensing data has shown temperature anomalies across the central
GOC between 2012 and 2017, with modest increases in the Midriff Islands Region, and
CTD data has also indicated that the observed subsurface warming could, in part, be
attributed to the presence of increasingly tropical Gulf of California Water between 2010
and 2013 (*Frawley et al., 2019*). In the Guaymas Basin it appears that there have been
several intrusions of tropical surface water into the GOC, a process which also transports
zooplankton of tropical affinity northward and that typically results in a decrease of the
zooplankton assemblage (*Gilly et al., 2022*). The same phenomenon has been observed
in the Pacific coast of the Baja California peninsula (*Durazo et al., 2017*). A phase shift
in multidecadal oceanic expressions could partially explain two strong El Niño events in
2009–2010 and 2014–2015, followed by weak La Niña conditions (*Frawley et al., 2019*).

Among the biggest ecological changes as a result of the increase temperatures in the
GOC has been the shift of the jumbo squid into its small phenotype. Prior to 2010, this
species migrated between the Baja California peninsula during the warm season and
Sonora during the cool season, this allowed the squid to forage year-round in productive
areas, which in turn supported its ability to live for up to 1.5 years or more, achieving
a large body size before maturing, spawning, and dying (*Gilly et al., 2022*). This pattern
was disrupted by a strong El Niño in 2009–2010 that led to the precocious spawning at

very small size, representing a switch to the phenotype characteristic of the species in the tropical, equatorial portions of its range (*Hoving et al., 2013*), where an early maturation is induced because of high temperature and low food availability (*Keyl et al., 2008*).

Though electronic tags on jumbo squid, it has been demonstrated that they spend little time at temperatures above 22 °C, therefore the depth of the 22 °C isotherm is of particular interest to their ecology, as they forage in the upper 75 m of the water column at night. If the 22 °C isotherm reaches such depth during the warm season it creates a stressful habitat and may impair foraging (*Gilly et al., 2022*). Prior to 2009, the 20 °C isotherm was at 50 m depth, but after de 2009-2010 it increased to ~75 m and remained there until the 2015–2016 El Niño which further depressed to 100 m, although later all isotherm slowly recovered, temperatures remained above those observed before 2010 (*Gilly et al., 2022*).

The 2009–2010 El Niño event produce a collapse of jumbo squid landings, due to the proliferation of the small-size phenotype (<40 cm mantle length ML). But unlike with previous El Niño events, there was little subsequent recovery, it was until 2014 that the historical average (~60cm ML) was reach, although for only a brief moment, as a positive shift in the Pacific Decadal Oscillation (PDO) and the emergence of a new El Niño produced a new collapse, and as 2022 hasn't recovered and there is doubt about if it will appear again in the GOC (*Frawley et al., 2019*; *Gilly et al., 2022*). The jumbo squid large phenotype started to disappear from the GOC around 2010 and were completely gone by 2015 and appears to have become firmly fixed in the region until present (2022; *Gilly et al., 2022*).

Jumbo squid fisheries in Sonora are based on the port of Guaymas, Sonora and are composed of an artisanal fleet, consisting of small boats called pangas (~7 m), and an industrial fleet of modified shrimp trawlers (10–15 m; *Cruz-Escalona et al., 2024*). Their fishing effort grew between 1996 to 2011, specially in the artisanal fleet with more than 1,600 boats in 2011 (*Cruz-Escalona et al., 2024*). The artisanal fleet, due to their technical capabilities can fish around the Guaymas Basin, south of the Midriff Islands Region, but the industrial fleet has the capacity to fish all around the GOC, including off the Baja California Sur coast (*Cruz-Escalona et al., 2024*). This implies that Sonora's jumbo squid landings used in the present study not necessarily are representatives of the jumbo squid present in the Midriff Islands Region. Nevertheless, *Frawley et al. (2019)* and *Gilly et al. (2022)* both found a positive linear relationship between mean mantle length (cm) and annual jumbo squid fishery landings (metric tons) in the Gulf of California. This indicates that jumbo squid landings are a useful indicator of phenotypic changes of the jumbo squid in the GOC. Which according to the same authors, is a phenomenon that has occurred all over the GOC, including the Midriff Island Region, therefore we choose to use this data, as the best available indicator of the status of the main prey of the sperm whales in our area of study. Although is important to recognize its limitations.

Our results, indicate that there is a good nonlinear correlation between sperm whales' sights per unit of effort and jumbo squid landings in Sonora ($p = 0.003$, SPU/ tons, *adj r*$^2 = 0.644$), which also drastically declined from 26,916 tons in 2009 to 139 tons in 2018. Signaling that although the jumbo squid can still be found in the GOC, its shift into the small phenotype not only collapsed its fisheries but also had a significant effect on the

presence of its predator, the sperm whale. The last sights of sperm whales in the Midriff Islands Region of the GOC occurred in 2015, when the jumbo squid large phenotype disappear (*Gilly et al., 2022*).

The disappearance of the sperm whales from GOC seems to occur beyond the Midriff Islands, as there have not been any reports of sperm whales in the Guaymas basin as well (*Gilly et al., 2022*). And the environmental and ecological changes in the GOC have also impacted other marine mammals, such is the case of the California sea lion (*Zalophus californianus*), which have shown a population decline in the GOC of 65% between 1991 and 2019 and that can be explained by the sustained warming of the GOC (*Adame et al., 2020*). One of the may preys of California sea lions, sardines, also had a fishery collapse after the 2009–2010 El Niño, with landings reaching levels close to zero by 2014–2015 (*Gilly et al., 2022*) as sardines are sensitive to thermal variation (*Petatán-Ramírez et al., 2019*).

The results obtained by the present and previous studies have shown that, prior to the warming period, the GOC and particularly the Midriff Islands Region was an important site for sperm whales, as demonstrated by the number of photo-identified individuals and population estimates. *Ruvalcaba Márquez (2013)*, though non-systematic surveys and data from *Centro Interdisciplinario de Ciencias Marinas* of the *Instituto Politécnico Nacional* (CICIMAR-IPN) and Dalhousie University, covered the Midriff Islands Region between 1993 and 2010, although his exact study area changed from season to season. He reported 2,191 photo-identifications corresponding to 589 unique individuals, with a high level of season-to-season variation in the number of photo-identifications, with the lowest number of one in 1992 and a maximum of 701 in 2002.

Our photo identification results oscillate between a maximum of 59 unique individuals in the 2010–2011 year and a minimum of five in the 2014–2015 year, follow by no sightings between 2015 and 2018. Although *Ruvalcaba Márquez (2013)* and the present study had different survey efforts, together they represent a 27-year timeline of sperm whale photo-identification in the Midriff Islands Region of the GOC.

Our sperm whale super population estimate for the Midriff Islands Region between 2009 and 2015 was, $\hat{N} = 354(\pm 95\%, SE = 94.7)$, which is lower than previous population estimates for the GOC. The super population for the 1986-2000 period has been estimated at 503 individuals (*Gerrodette & Palacios, 1996*), $\hat{N} = 941(95\%, SD = 303, LCI = 386, UIC = 2,493)$ for the 1998–1999 period (*Jaquet, Gendron & Coakes, 2003*), and $\hat{N} = 1,070(95\%, LCI = 734, UCI = 1,623)$ for an undisclosed period (*Ruiz, Ramírez & Bracho, 2006*). While the lower estimation reported by the present study could be attributable to its smaller study area, it could also indicate a downward trend in the sperm whale population in the GOC, especially given the positive warming trend and ecological changes that have been reported for the last three decades.

## CONCLUSIONS

The present study has shown the importance of the use of long-term databases to estimate the population trend of a long-lived odontocete species. However, detailed analysis is required to fully understand the population trend of the sperm whale and its individual

movements along the Gulf of California (GOC). A simple comparison of previously published population estimations reveals that, although those made for the study area are inconsistent, there seems to be a downward trend in the population of both sperm whales (*Gerrodette & Palacios, 1996*; *Jaquet, Gendron & Coakes, 2003*; *Ruiz, Ramírez & Bracho, 2006*) and their main prey, the jumbo squid (*Frawley et al., 2019*; *Hoving et al., 2019*), in the central GOC. Overall, the results of the present study suggest a close relationship between sperm whales and jumbo squid, at least in the region of interest. While jumbo squid landings tend to decline and recovery after an *El Niño* event, in more recent years they do not seemed to have recovered, and the small phenotype of the jumbo squid seems to fix (*Frawley et al., 2019*; *Gilly et al., 2022*). Both the departure of sperm whales from the study area and the switch of jumbo squid into their small phenotype seem to be the result of environmental and ecological changes (*Gilly et al., 2022*) that are occurring along the entire GOC (*Adame et al., 2020*). A sustained increase in warmer waters since the 1930's, strong El Niño events and positive PDO phases in the last 15 years (*Frawley et al., 2019*; *Gilly et al., 2022*; *NCEI, 2024*) are possible producing a ''tropicalization'' of the GOC (*Frawley et al., 2019*), with the subsequent decline of keystone species like jumbo squid (*Frawley et al., 2019*; *Gilly et al., 2022*), as well as an effect on sentinel marine mammals such as the sperm whales.

## ACKNOWLEDGEMENTS

This work was conducted as part of the Marine Mammal Program of the Prescott College Kino Bay Center for Cultural and Ecological Studies, based in Sonora, Mexico. We want to thank Lorayne Meltzer, Director of the Kino Bay Center, for all of the support and trust received over these years of work. We also wish to thank our boat captain Cosme Damian Becerra for the personal bond established with us and his professionalism in ensuring our safety at sea. Thanks to Dr. J. Urbán-Ramírez, of the *Programa de Investigación de Mamíferos Marinos* (PRIMMA or Marine Mammal Research Program) at the *Universidad Autónoma de Baja California Sur*, for his support in enabling us to work under his scientific research permits. Benjamin Stewart for proofreading the manuscript.

### Funding

This work was supported by The Lucile and Packard Foundation and Prescott College Inc. The Fundación Universidad de las Américas Puebla covered publishing fees and scholarship for Daniela Ahuatzin Gallardo. The funders had no role in study design, data collection and analysis, decision to publish, or preparation of the manuscript.

### Competing Interests

The authors declare there are no competing interests.

## Author Contributions

- Héctor Pérez-Puig conceived and designed the experiments, performed the experiments, prepared figures and/or tables, authored or reviewed drafts of the article, and approved the final draft.
- Alejandro Arias Del Razo conceived and designed the experiments, analyzed the data, prepared figures and/or tables, authored or reviewed drafts of the article, and approved the final draft.
- Daniela Ahuatzin Gallardo conceived and designed the experiments, performed the experiments, analyzed the data, prepared figures and/or tables, and approved the final draft.
- Jaime Bolaños analyzed the data, authored or reviewed drafts of the article, and approved the final draft.

## Animal Ethics

The following information was supplied relating to ethical approvals (*i.e.*, approving body and any reference numbers):

Secretaría de Medio Ambiente y Recursos Naturales (SEMARNAT), Dirección General de Vida Silvestre <General Directorate of Wildlife> provided full approval for this research SGPA/DGVS/01329/12

## Field Study Permissions

The following information was supplied relating to field study approvals (i.e., approving body and any reference numbers):

Secretaría de Medio Ambiente y Recursos Naturales (SEMARNAT), Dirección General de Vida Silvestre <General Directorate of Wildlife> provided full approval for this research

## Data Availability

The raw data is available in the Supplemental Files.

## Supplemental Information

Supplemental information for this article can be found online at http://dx.doi.org/10.7717/peerj.18117#supplemental-information.

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
