# Peer review of "The departure of sperm whales (Physeter macrocephalus) in response to the declining jumbo squid (Dosidicus gigas) population in the central portion of the Gulf of California"

_PeerJ, doi:10.7717/peerj.18117_

## Round 0.1 · original submission · Major Revisions

The reviewers and I appreciate the work you've put into this analysis. The reviewers have highlighted a number of areas where the manuscript could be improved -- the majority of these being clarifications, or additional details that should be straightforward to address.

Reviewer 1 ·

Basic reporting

This work represents an advancement in the knowledge of the sperm whale population in the Gulf of California, providing data to understand recent trends of population and potential driving factors.

Is well written in a professional manner, so no problems in this regard.

The references seem to provide enough background information, covering the previous knowledge of the species in the region.

The structure of the text seems right, easy to read, with pieces of information in the right order.

Experimental design

In line 93 you state that the study is aimed at estimating the abundance of sperm whales in the eastern Midriff Islands region, as well as try understanding the influence on abundance of the availability of its main prey in the region. You use Figure 1 to depict the Midriff Islands region. Later, in the methods section, between lines 119 and 133, you describe field sampling procedures, saying that surveys occurred in weekly periods, using small boats at speeds about 7 kn. In the Results section, you say that weekly surveys had an average duration of 6.5 hours. Although it is not made explicit, we can assume that, on every survey, you departed from Bahia Kino and returned to the same port.

Considering the average speed and duration of surveys, it can be calculated that, in average, you were able to navigate by a total of about 84 kilometers, which means, given that you needed to get back to port, that you were able to get away from the port a maximum, in average, of 42 kilometers. However, you were searching for whales, and stopping during sightings, hence the autonomy would be reduced even more. As such, in my opinion, the methodological description is incomplete, needing to depict which was exactly your study area, that seems to be a relatively small polygon near to Bahia Kino and Tiburon Island. Also, you need to say if the coverage polygons varied significantly between weekly surveys.

I suggest you add a polygon in Figure 1 showing your actual study area coverage and specify in the text how did you select the navigation routes between surveys. Did you select the starting navigation route, once departing from harbor in a random way, or following some belief about the distribution of whales. This needs to be specified to understand potential biases, which is not problematic if they were introduced systematically throughout the study period.

Besides this, you mentioned that used databases of CONAPESCA of the State of Sonora about squid landings. I suppose this information covers all the areas where fishery operates in the Gulf of California, not only in your study area. I suggest that you consider or discuss if these databases represent the abundance or availability of squid in your study area, or if there are problematic geographic variations on catches, that need to be managed to avoid biases. In addition to, it would be better to express catches in terms of CPUE, as you did with sperm whale sightings, as landings could not capture the size of the operational fleet and other factors, which could make the parameter not representative of the abundance or availability of squid.

Validity of the findings

According to results presented in Table 2, about annual abundance estimates, it seems that population was stable throughout the study period, even with a potential increase, until not more sightings occurred from 2016. However, in Table 3, it is shown that CPUE of sperm whales (I would say “sightings per unit effort”) have the opposite behavior, with a steep decreasing pattern. This is contradictory and would result in different interpretations depending on the dataset used. It is, in my opinion, intuitive that CPUE of sperm whale sightings must be proportional to abundance, unless methodological procedures could be inserting some source of unknow bias. I would suggest a review of the filed protocols, to confirm that no significant variation between weekly surveys could be introducing biases. On the other hand, review the abundance estimation model, checking out the validity of assumptions on the model parameters used.

On the other hand, the squid landings seem to have two periods, first with a steep decrease among 2009 and 2012, and a second with a kind of stability, among 2012 and 2018. That is why, in my opinion, Figure 3 looks as two aggregates of data, one under the stability period and the other during the decreasing period, with remarkable differences in CPUE of sperm whales’ sightings. The aggregate in the stability zone looks like highly variable. Your adjusted linear model, shown in Figure 3, estimates that at zero squid landings, it is expected to still have some sperm whale sightings, which departs from the trend shown in the data, with zero sightings, for three years, under short landings. I think you must explore the use of alternative non-linear models, like generalized linear models (GLMs), that better fit the trend in your data.

Additional comments

In my opinion it is worth to accept the document for publication, but needs a major revision, reviewing the next aspects:

• Make a complete description of your field methods, specifying your study area extent and its potential variation between weekly surveys. Also, describe how the navigation routes were selected, by random procedures, beliefs about distribution animals, systematic first track of navigation, or any other.
• Discuss if the study area is representative of the Gulf of California or the Midriff Islands region. If not, state that your results are representative only of your study area or, as you say, of the eastern Midriff Islands region, specifying what “eastern” means in terms of geographic extent.
• Try to construct a CPUE of squid catches from the databases available from CONAPESCA to be used in further modelling. If it is not feasible, discuss if landings are reasonably proportional to CPUE, based on data on fleet size between years or similar data.
• Consider if squid landings database is representative of your study area, or if some biases could be inserted. Has the fishing fleet used similar fishing grounds over the years? Is there fishing effort inside your study area, and this is relatively constant over years, or it is variable as in other regions where Sonora fishing fleet used to do the activity. In my opinion this kind of questions could help to analyze and determine if the landings dataset represent your study area.
• Review your abundance estimation model, as the estimated yearly abundances do not seem to agree with your data on CPUE of sperm whale sightings. Or discuss why both datasets could differ and still be valid.
• Consider using a non-linear model to analyze the relationship between squid and sperm whale abundances (expressed as CPUE, landings, or any other parameter).

·

Basic reporting

The study investigates the correlation between the sperm whale (Physetur macrocephalus) and the decline of the giant squid (Dosidicus gigas) population in the central portion of the Gulf of California. The topic is of high interest either for the conservation of the target species but also for their role as indicators of the status of marine biodiversity and the ecological process in this key marine area.
However, prior to be accepted for publication, the MS needs some revisions.
A main point regards the clear identification and description of the objective of the study. Several studies were already conducted in the GOC on the same topic, which are correctly included in the introduction (line 41-90): a better identification of the need for this further investigation and, consequently, on the novelty of this present study is required.
On this regard, a knowledge gap was identified in line 91 referring to the fact that ‘no studies have focused on either the progressive variation in sightings or the size of the sperm whale population in the GOC’. As it is however, the sentence looks unclear and there are some imprecisions in the relate description of the objectives of the study.
In general, I suggest revising the terminology for the lines 90-98 and the whole MS such for the examples below:
- consider using the term ‘trend’ instead of ‘progressive variation’;
- the term ‘sightings’ in line 91 doesn’t include reference to the survey condition (e.g., area covered by the effort, time spent in observation and so on) so could be misleading of a variation either due to the species presence or to different effort conditions;
- the term ‘population’ needs to be clarified at the beginning as in this MS it is used to indicate the local individuals visiting the study area.
- The use of the term ‘population dynamic’ could be a bit confusing here because, even if the population dynamic is a branch of the ecology dealing with spatio-temporal variations of population/s, it mainly focuses on changes due to birth or death rates, immigration and emigration, so not really the case of this study. I suggest changing the terminology referring for example to trends in the number of sperm whales visiting the study area in the GOC (..and its relationship with the abundance of its main prey).

Experimental design

A second point regards the clear definition of the time scale of the investigation within which the study was conducted and the analyses performed. There are many terms that refer to season, intra-annual interval, intra-season,….within the MS. For example, in line 94 the authors refer to ‘seasonal variation’, but it is unclear what is intended here. I couldn’t find in the study a result about variability among different seasons. Was the effort performed in a specific season? So that the yearly comparison is based on comparing the same season among years? In line 137 it is mentioned the photo-recaptures for the ‘intra-season period’ but it is unclear which season/s was/were surveyed. The same for the results based on multiple or intra-annually season (e.g. line 161-193). Please define at the beginning of the method which season (or seasons) was/were surveyed in which year. Maintain consistency in the terminology throughout the MS.
A third important point regards the method used to calculate the CPUE (and so the trend among years) and, in particular, for defining the effort on which normalize the number of sightings. For example, in line 124 and 127 is stated that the effort for calculating the CPUE was measured in hour: is it possible that difference in vessel speed or in the number and length of the stops could have influenced the sightings’ probability (e.g., more or less area covered in search)? Also, was the time calculated excluding the time spent with the animals after the first sight or was this time included? Please consider the potential influence of the search effort on the CPUE annual results, or evaluate the possibility to calculate the effort as km travelled in search.

Validity of the findings

From the results, it seems that the main output is the correlation between the sperm whale abundance indexes calculate as CPUE and the giant squid landings, while the population estimates does not show any trend before the disappearance of the last three years. I suggest better commenting on this aspect in the discussion. As for example, it seems that, despite the declining in CPUEs during the years, a variable number of animals visited the study area. Perhaps for shorter period? So that the probability of encounter was lower each year? Could be correlate with the number of individual recaptured?

Additional comments

Oher minor suggestions are listed below:
Line 101-117 Study area: I suggest inserting all the toponyms mentioned in the MS in figure 1 to facilitate the understanding also for people not familiar with the study area.
Line 130: was the geographical location taken at first sight? Please specify.
Line 141 ‘by season’ : which season? See above the comment on the season.
Line 158-159: the CPUE is mentioned only here but there is no clear specification about the method to calculate this index, while it is reported in the results line 186-187. I suggest moving this information in the methods session. Moreover, as the main output is based on the decline of the CPUE used as index of abundance of individuals visiting the study area and its correlation with the giant squid landings data, I’d suggest to better clarify this index and the method used to calculate it (as for the example of the method used to calculate the effort as mentioned previously).
Line 163-164: there is a bit of confusion between number of sightings and number of individual. Here are reported 73 sperm whale sightings of 648 individuals, while in table 3 is indicated the number of individuals per year as ‘Sperm whale sightings’. Consider modify the heading of table 3 as ‘Number of individuals (or speciments)’ or ‘number of sperm whales sighted’
Line 208-212: unclear, please rephrase it.
Line 213-216: in general, it is not easy to follow the comparison between the two studies (Ruvalcaba-Marquez 2013 and the present study) and it seems that difference where quite remarkable. Perhaps a clearer figure from which extrapolate the overlap or differences in the study area locations could help.
Line 217-229: see previous comment on the findings of the study.
Figure 1 please insert the toponyms mentioned in the text
Figure 2 I imagine these are the number of sperm whales sighted (not the number of sightings)
Figure 3 Y axes insert the formula of the CPUE
Table 1 there is no clear mention in the method on what is intended for intra-annual or inter-annual (see previews comment on season/s)
Table 3 :
- insert the reference for the squid landing data (eg. CONAPESCA DB 2016).
- Correct the heading on the number individual of sperm whale sighted

---

## Round 0.2 · Minor Revisions

I appreciate the work you've put into the revision, and think it's much improved. I think the additional details, and new methodology (GAMs) are a good choice. I have 2 additional requests. First, the methods section needs additional citations -- GAMs, citing the R software, etc all need to be cited. You can use citation() in R, or citation("mgcv") for examples. Second, please link to the code in a publicly accessible repository (like Github) where readers may see the code to replicate the analysis. Both of these changes will help readers better understand the analysis.

---

## Round 0.3 · accepted · Accept

Thanks for adding the additional citations, and uploading code. Congratulations!